**Data Availability Statement:** All relevant data are within the manuscript and its Supporting information files.

**Funding:** This work is gratefully supported by Rowland Fellowship (S.C.H), sponsored research

# Sensitizing drug-resistant cancer cells from blood using microfluidic electroporator

**Hyun Woo Sung**[1], **Sung-Eun Choi**[2], **Chris H. Chu**[3], **Mengxing Ouyang**[4], **Srivathsan Kalyan**[2], **Nathan Scott**[2], **Soojung Claire Hur**[2,5,6]*

1 Department of Chemical and Biomolecular Engineering, Johns Hopkins University, Baltimore, Maryland, United States of America, 2 Department of Mechanical Engineering, Johns Hopkins University, Baltimore, Maryland, United States of America, 3 Department of Internal Medicine, Virginia Mason Medical Center, Seattle, Washington, United States of America, 4 Department of Bioengineering, University of California, Los Angeles, California, United States of America, 5 The Sidney Kimmel Comprehensive Cancer Center, Johns Hopkins Hospital, Baltimore, Maryland, United States of America, 6 Institute of NanoBioTechnology, Johns Hopkins University, Baltimore, Maryland, United States of America

* schur@jhu.edu

## Abstract

Direct assessment of patient samples holds unprecedented potential in the treatment of cancer. Circulating tumor cells (CTCs) in liquid biopsies are a rapidly evolving source of primary cells in the clinic and are ideal candidates for functional assays to uncover real-time tumor information in real-time. However, a lack of routines allowing direct and active interrogation of CTCs directly from liquid biopsy samples represents a bottleneck for the translational use of liquid biopsies in clinical settings. To address this, we present a workflow for using a microfluidic vortex-assisted electroporation system designed for the functional assessment of CTCs purified from blood. Validation of this approach was assessed through drug response assays on wild-type (HCC827 wt) and gefitinib-resistant (HCC827 GR6) non-small cell lung cancer (NSCLC) cells. HCC827 cells trapped within microscale vortices were electroporated to sequentially deliver drug agents into the cytosol. Electroporation conditions facilitating multi-agent delivery were characterized for both cell lines using an automatic single-cell image fluorescence intensity algorithm. HCC827 GR6 cells spiked into the blood to emulate drug-resistant CTCs were able to be collected with high purity, demonstrating the ability of the device to minimize background cell impact for downstream sensitive cell assays. Using our proposed workflow, drug agent combinations to restore gefitinib sensitivity reflected the anticipated cytotoxic response. Taken together, these results represent a microfluidics multi-drug screening panel workflow that can enable functional interrogation of patient CTCs *in situ*, thereby accelerating the clinical standardization of liquid biopsies.

## Introduction

Liquid biopsies are rapidly emerging as a paradigm shift in clinical cancer treatment for their potential in non-invasive, longitudinal surveillance of patient disease prognosis [1–3]. Liquid biopsies can be performed using bodily fluid samples, ranging from pleural effusions,

agreement between Harvard University and Vortex Biosciences, Inc. (S.C.H), the Susan G. Komen Career Catalyst Award (S.C.H) under award number CCR19609203, and NCI of National Institute of Health (S.C.H) under award number R21CA229024. The funders had no role in study design, data collection and analysis, decision to publish, or preparation of the manuscript.

**Competing interests:** S.C.H. benefits financially from royalty payments from the Vortex Biosciences, Inc. This does not alter our adherence to PLOS ONE policies on sharing data and materials.

cerebrospinal fluid, urine, and blood, thereby representing a wide array of tumor-associated information that can be measured for tracking the progression of the disease [4]. The valuable analytics within liquid biopsy samples include intact cells, cell-free nucleic acid, and extracellular vesicles shed directly by tumors to colonize multiple secondary tumor sites. As an alternative source for primary cells, liquid biopsies overcome several key limitations of traditional solid tumor biopsies that necessitate this shift. First, depending on the anatomical location or inadequate size of the tumor, surgical biopsies isolate limited selections of the tumor and place a significant burden on the patients themselves. Second, liquid biopsies encompass a broader pool of malignant cells in circulation shed by the primary and secondary tumors and therefore reflect a more comprehensive cross-section of heterogenetic tumor properties. More importantly, solid biopsies reference a temporal window at the time of sampling, which may factor out important changes in the arc of disease progression. As tumors have been observed to exhibit a highly dynamic molecular landscape [5–7], longitudinal monitoring of tumor heterogeneity, genomic evolution, and disease progression are pivotal for making informed clinical decisions and directing appropriate therapeutic avenues during the course of treatment.

For their functional role in orchestrating the metastatic cascade, CTCs found in peripheral blood samples are one of the cornerstones for liquid biopsy analysis [8, 9]. CTCs are cancer cells shed directly from primary and secondary tumors into the circulatory system and are well-documented to be harbingers of the metastatic cascade [10, 11]. During the first 24 hours of extravasation, CTCs represent comprehensive snapshots of real-time tumor progression [12, 13], providing an instrumental window for tracking the tumor state in real-time. However, clinical practices using liquid biopsies are limited because CTCs are obscured from the high background from billions of blood cells (1–100 CTCs per 1 mL whole blood [14, 15]). Purification challenges exacerbate the establishment of standardized operating procedures and workflows and thus remain the principle roadblock for adapting routine patient liquid biopsy testing in the clinic.

Advancements in cell enrichment techniques have generated a plethora of data detailing the molecular and functional characteristics of CTCs. In response to treatment for multiple types of solid cancers, expression patterns of CTC biomarkers shift dynamically which suggests that CTC biomarker analysis can be an effective measure of therapeutic response [16–19]. This is particularly significant because through epigenetic mutations and selective pressure from treatment, tumor cells that are in a constant state of genetic evolution can develop subclones that are resistant to front-line therapies [7, 20–22]. Because these cells represent a live temporal window, stable cell lines of CTCs have been generated *in vitro* for tumor characterization applications. These cell lines can further be developed into functional models for drug susceptibility testing against a library of chemotherapeutic agents or to recapitulate tumorigenic behavior in xenografted mice models [23–28]. Limited success rate and prolonged processing time to establish these cell lines from isolated CTCs hinder their overall clinical translation, as culture passaging further induces phenotype and genotype altercations in derived cell lines that may not accurately depict the native tumor physiological state. Nonetheless, these critical results demonstrate that short-term drug screening and functional evaluation of CTCs are possible. Despite limited compatibility with liquid biopsy samples due to their rare cell count, existing drug-screening platforms utilizing patient tumor cells [29–31] also underscore this necessity in probing primary cell populations for identifying tailored therapeutic options. As a valuable primary cell source, systems capable of directly assaying CTCs in patient liquid biopsy samples for functional interrogation are essential to accelerate the clinical adaptation of liquid biopsies.

We have previously reported platforms trapping viable cells in suspension with an electroporation-based transfection system for delivering combinations of exogenous biomolecules at

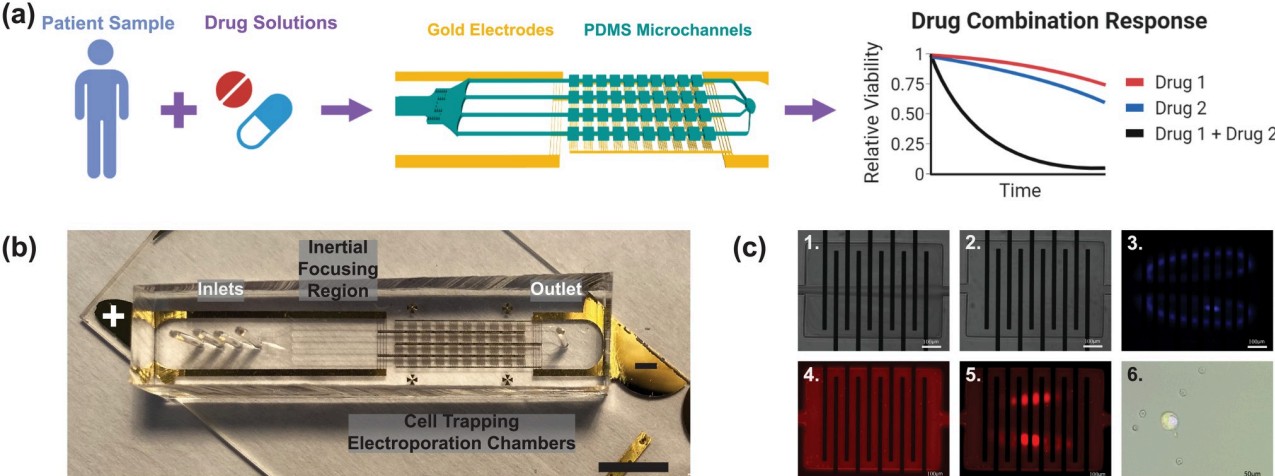

**Fig 1. Overview of workflow for assessing drug response from cancer cells.** (a) Schematic workflow for directly assaying diseased cells from patient liquid biopsy samples for subsequent drug combination screening. Cell-trapping PDMS microfluidic chambers (turquoise) and Au electrode arrays (gold) of the assembled device allow cell purification from blood and subsequent drug response analysis (b) Overhead photograph of the device. The device consists of an inlet region accommodating multiple solution ports with coarse filter for blocking debris, inertial focusing region for size-based focusing of larger cancer cells from smaller blood-borne contaminant cells, and cell trapping region (10x4 chamber arrangements) fitted with patterned microelectrodes (10 planar microarrays per chamber) for cytosolic delivery of bioreagents via electroporation. Scale bar represents 5 mm. (c) Representative time-lapse panel images of the integrated process consisting of 1. cell trapping, 2. flushing away blood cells, 3. visualization of vortices, 4. molecular incubation, 5. visual confirmation of successful electroporation, and 6. merged microscopic image of both background blood cell (white arrow) and a viable, electroporated cancer cell (black arrow), indicated by overlapping red (Propidium Iodide) and green (Calcein AM) fluorescent signals. Scale bar represents 100 μm for panels 1. - 5. and 50 μm for panel 6.

controlled quantities [32, 33]. Vortex-based rare cell isolation techniques for target cell separation using size-based biophysical markers have enabled both enrichment and collection of intact cells. Electroporation of these trapped cells reversibly permeates the cell membrane and precisely mediates the delivery of a vast multitude of biomolecules such as drugs, ribonucleic acids (RNAs), and plasmids. Furthermore, microscale electroporation is particularly useful when documenting responses from rare and sensitive cells to combinatorial anticancer therapeutic agents, which could have vastly contrasting cell membrane transport properties. Additionally, on-chip cell trapping, sequential electroporation, and controlled release of concentrated cells eliminates various pipetting steps typically needed to perform functional assays involving multiple reagents, thereby eliminating issues with retaining rare cell counts. Utilizing this coupled system, we report an extended electroporation workflow that processes complex body fluid for direct assessment of viable target cells. To demonstrate its clinical applicability, we describe a routine for performing drug response assay onto cancer cells spiked into blood samples. As proof of concept for clinical translation using patient blood samples, we detail a procedure using drug-resistant cell line HCC827 GR6 spiked into the blood for subsequent enrichment and drug response screening. We envision our system as a powerful workflow for processing valuable information from viable CTCs to complement the utility of liquid biopsies in guiding clinical therapeutic decisions (Fig 1a).

## Materials and methods

### Chip fabrication

The channel geometry and operational procedure of the device are adapted from our previously reported vortex-assisted cell electroporation system [32]. The microfluidic

electroporation device was designed using AutoCAD (Autodesk, Inc., USA), and the microfluidic channel and electrode photomasks were printed and purchased from Photo-Sciences (Photo Sciences Inc., USA). The gold electrodes on plain glass slides (VWR, USA) were patterned by spin coating positive photoresist (SPR 220–3, Rohn Haas, USA) layer. SPR 220–3 was spin-coated onto the glass slides at 2000 rpm for 50 seconds ramping at a rate of 100 rpm/s. The glass slides were then baked on a 115 ˚C hot plate for 5 minutes, then on a 65 ˚C hot plate for 3 minutes. The electrode pattern was then fabricated on these glass slides by following conventional photolithography procedures (EV620) and developed under CD-26 developer (Microposit CD-26, Rohn Haas, USA) for 60 seconds followed by immersion in DI water. Electrodes were deposited using an electron beam deposition tool (Sharon Vacuum CVS-6 Evaporator, USA) at a vacuum environment of $6^{e-6}$ torr and depositing 10 nm thick chromium and 300 nm thick gold layers sequentially (Kurt J Lesker, USA). The sacrificial layers of electrodes were lifted off by sonication in an acetone bath for 5 minutes.

The casting mold for the microfluidic channel was fabricated using a negative photoresist (KMPR 1050, Microchem, USA) following standard photolithography procedures. Targeted heights (70–80 μm) of the fabricated microstructures were validated using a surface profilometer (Keyence VK X250). Poly(dimethylsiloxane) (PDMS; Sylgard 184 silicone elastomer kit, Dow Corning, USA) replicas were generated using a 5:1 ratio of PDMS and curing agent. PDMS was degassed for 1 hr and cured at room temperature for 48 hrs. Cured PDMS replicas were removed from the mold, and solution injections and outlet holes were created by using a pin vise (Pin vise set A, Syneo). Microfluidic channels and the fabricated electrode glass slides were treated with oxygen plasma (Technics MicroRIE, USA), aligned, and irreversibly bonded to the electrodes onto the electrodes (Fig 1b).

## Device operation

Cells in suspension are inertially focused, trapped, and contained within micro vortices formed inside the trapping chambers and subsequently electroporated for cytosolic delivery. Electroporation is visualized in real-time by the delivery of cell-impermeable dye, allowing for rapid modification of electroporation parameters to optimize electroporation performance (Fig 1c). Real-time visualization of fluorescent orbit traces from electroporated cells is shown in S1 Fig. The trapping region was designed to select larger (>10 μm[1]) cancer cells from smaller blood cells (2–10 μm [34–36]).

## Cell line and culture

The epidermal growth factor receptor (EGFR)-mutant NSCLC cell line, HCC827, and gefitinib resistant line, HCC827 GR6, were kindly provided by Dr. Pasi A. Jänne's group at the Dana-Farber Cancer Institute. While HCC827 wt cells demonstrate sensitivity to gefitinib, a tyrosine kinase inhibitor (TKI) targeting EGFR, HCC827 GR6 cells are resistant to gefitinib. PHA-665752, a c-MET inhibitor, restores gefitinib sensitivity in HCC827 GR6 cells [37]. The growth media for both HCC827 cell lines were composed of Roswell Park Memorial Institute 1640 (RPMI 1640, Gibco®, Life technologies, USA) supplemented with 10% (v/v) heat-inactivated fetal bovine serum (HI-FBS, Gibco®, Life technologies, USA) and 1% penicillin-streptomycin (Sigma-Aldrich Co., USA). HCC827 GR6 cells were cultured in 1 μM gefitinib every ten passages to selectively screen drug resistance colonies. Cells were passaged at 70% confluency and subcultured into T-75 flasks (VWR, USA) in 10 mL of growth media at a ratio of 1:5. Cells were grown in a humidified incubator according to the supplier's protocol at 37˚C in a humidified 5% $CO_2$ environment. Media was exchanged to fresh complete media bi-weekly. Cells

were harvested by treating with 0.25% trypsin-EDTA (Gibco®, Life technologies, USA) for 3 minutes.

## Cell suspension preparation

Cells were pelleted by centrifugation at 1100 rpm for 5 min and resuspended into 2 mL fresh growth media before passing the suspension through cell strainers (40 µm Scienceware FlowMi, Bel-Art, USA) to remove external debris that may clog the microchannels within the device. Cell suspensions were diluted in fresh media to have a final concentration of $5 \times 10^4$ cells/mL for electroporation experiments. For fluorescent-based electroporation experiments, cells were stained with the nucleus indicator NucBlue™ Live ReadyProbes™ Reagent (Hoechst 33342, Invitrogen, USA) following manufacturer protocol prior to electroporation.

## Solution exchange

Similar to a previously described pneumatic flow control system [32, 33, 38], a custom-built electronic pneumatic flow control unit was used to regulate pressurization of nitrogen gas (AirGas, USA) into control solution vials containing DPBS wash buffer (HyClone, Cytiva, USA) and drug solutions. Briefly, the system consists of 4 electro-pneumatic regulators (ITV0030-3UBS, SMC Pneumatics, only three were used for the current study) controlled by custom-written LabView software. 40 psi (equivalent to about 1.0 mL/min flow rate) and five psi were used for injecting pressure and idle pressure, respectively, for these solution vials to minimize pressure buildup and subsequent backflow back into the solution vials. The solution vials were linked to the inlet ports of the PDMS chip via PEEK tubings (1/32" OD, 0.020" ID, IDEX, USA) and were fitted with inline check valves (IDEX, USA) to further impede backflow.

Prepared cell suspensions both in media and blood were injected into the microfluidic device with Standard Infuse/Withdraw Syringe Pump (Harvard Apparatus). For the cell suspension PEEK tubing into the inlet port, a manual shut-off valve assembly (IDEX, USA) was fitted instead of the inline check valve to block backflow. The manual shut-off valve was opened during the cell trapping step and shut promptly after. Prior to the inflow of cell samples, stable trapping vortices were established in the device by flowing wash buffer solution at 40 psi for 30 seconds. For experiments with cell suspensions in media, the syringe pump was programmed to infuse at a rate of 1.2 mL/min for 50 seconds, followed by a withdrawal rate of 0.5 mL/min for 30 seconds. For experiments with cell suspensions in diluted whole blood, the syringe pump was programmed to infuse at a rate of 1.0 mL/min for 180 seconds, followed by a withdrawal rate of 0.5 mL/min for 30 seconds. Prior to solution exchange and electroporation, cells were briefly flushed with wash buffer at 40 psi for 60 seconds to remove cells not trapped within vortices from the entire device.

A transition dual-flow of solution vials at 40 psi for 10 s was done when switching between solutions, as abrupt on-off switching may fluctuate the flow rate into the device and adversely affect stability of the vortex. Switching between solutions this way ensures that the flow rate throughout the device, at any point, is sufficient to retain cell trapping.

## Electroporation procedure

Promptly after cell trapping, trains of electric pulses were applied to the electrode by using a compact and programmable custom-built AC square wave pulse generator. The standard electrical parameters used for all the experiments described in this manuscript were set as 10 square wave pulses with frequency, f = 20 kHz, pulse width, τ = 1ms, and pulse interval, Δt = 1 s. The electrical field in the microfluidic chamber was generated using a custom-built signal

generator. Briefly, a power supply (E3630A Triple Output DC, Hewlett Packard, USA) provided the desired voltage to a transistor H-Bridge. The square pulse, frequency, duration, pulse width, and interval were all controlled by a microprocessor (MSP430F5529, Texas Instruments, USA). Pulses were also tracked using an oscilloscope (Agilent, USA). The finite-element method (FEM) was used to simulate the electric field generated via electroporation (COMSOL Multiphysics v4.4, Electric Currents Package) [32]. Reported electric field intensities are from a height Z = 60 μm from the base of the electrodes.

Prior to cell collection, the device and containing trapped cells were briefly flushed with the neutral DPBS wash buffer solution (30 s) at 40 psi to primarily flush away residual reagents that may inadvertently be collected with the cells in the well-plate and interfere with accurate electroporation efficiency assessment. The processed cells were collected from the outlet PEEK tubing directly onto 96-well plates by lowering the wash buffer pressure from 40 psi to 5 psi for 10 s, then back to 40 psi for 5 s.

## Fluorescent microscope imaging

*in situ* observation and imaging of the electroporation process were monitored using an inverted microscope (Eclipse Ti, Nikon Inc., Japan) equipped with a mercury Epi-fluorescent light source (Intensilight, Nikon Inc., Japan), fluorescent filter cube sets, and a CCD camera (Clara, Andor, USA). Entire 96 well-plate fluorescent images were obtained using the 10x objective to obtain and stitch automated multi-point captures using in-house software (NIS-Elements, Nikon Inc., Japan).

## Quantitative analyses for electroporation efficiency

A combination of Fiji (ImageJ) and CellProfiler [39] software was used for determining electroporation efficiency in fluorescent molecule delivery experiments. For these experiments, trapped cells were electroporated and then incubated in membrane-impermeant nucleus dye solution for 40 seconds. After flushing with wash buffer, cells were collected onto clear 96-well plates and then incubated in Calcein acetoxymethyl (AM) for 20 min before centrifuging (180 g, 5 min) and resuspension in DPBS for whole-well fluorescent imaging (<1 hr).

A custom-built image analysis pipeline generated 2D fluorescent intensity maps from individual cells collected post-electroporation to assist with electroporation efficiency assessment. Gate intensity values for determining cell viability and successful electroporation were set using intensity values emitted from conventional well plate images of cell populations incubated in the same concentrations of fluorescent molecules. S2 Fig shows an example of the generated intensity maps along with applied input voltages. The low amount of collected cells (< 1k cells per run) precludes direct analysis of our collected samples with flow cytometry.

Briefly, coordinate locations of individual cells were mapped using Fiji to generate single-cell crops of individual cells using Python (Spyder-IDE, MIT). These batch crop images were processed for qualitative fluorescent intensity measurements using CellProfiler (CellProfiler, Broad Institute). Only cells with recognizable Hoechst signals (indicating the presence of a viable nucleus prior to electroporation) were analyzed for FITC and TRITC intensity measurements. Aberrantly recognized signals or images with overlapping cells were manually examined and removed. The background-subtracted intensity values from each cell were plotted on a 2D fluorescence intensity map. Threshold intensities for successful electroporation from nucleus-binding dye were defined to be greater than $I_{Mean}+2\sigma$, where $I_{Mean}$ and $\sigma$ are the average intensity and standard deviation from the live cell populations in the control group (i.e., fluorescent intensities of membrane-impermeant nucleus dye emitted by more than 95% of the live cells in the control group were lower than the threshold). Thresholds for cell viability

were defined as the maximum Calcein signals emitted from the dead cell populations in the control group to minimize false positives in the electroporation quadrant. Calculations of transfection efficiency, cell viability, and collected cell counts were calculated using formulas as previously described [32, 38].

### Drug preparation

Gefitinib (S1025) and PHA-665752 (S1070) were reconstituted in DMSO (Selleck Chem, USA). Reconstituted stock drug solutions were diluted in DPBS (>1000x) for electroporation experiments. For drug electroporation experiments, cells were electroporated and incubated in drug solutions with total drug solute flown during the duration matching control well plate experiments (i.e., 1, 0.2 μM for gefitinib and PHA-665752, respectively, in 100 μL total volume). Therefore, electroporation experiments were conducted by flowing solution vials containing gefitinib and PHA-665752 at concentrations of 100 and 20 nM, respectively, at 40 psi for 60 s each, either as single agents or sequentially for combination assessment, onto trapped HCC827 GR6 and wt cells.

### Cell viability measurements

Cell viability was reported using a non-lytic bioluminescence assay measuring the reducing potential of metabolically active cells [40]. Promptly after electroporation and incubation in drug solutions, cells were collected in wash buffer onto white opaque assay well plates and centrifuged (180 g, 5 min) using a well-plate centrifuge (LMC-3000, Grant). Collected cells were resuspended in complete RPMI media containing 1x RealTime-Glo™ MT Cell Viability assay reagent (Promega). The viability of cells post-electroporation was measured every 24 hours using Cytation 5 (BioTek, Agilent) plate reader over the course of 72 hours in a continuous read format. Due to the natural variance in collected cell count per electroporation run, fold-change for each drug condition was calculated by dividing luminescence readings from every 24 hrs to the initial 0 hr readings. This fold-change was compared to the fold-change observed from cells electroporated without drug incubation and reported as % viability. Minimal variance in the fold-change of intensity values were observed in calibration wells plated with a wide range of cells, suggesting that the cell density variation effects onto fold-change in cell chemiluminescence are negligible (S5 Fig).

### Spiked blood preparation

In order to demonstrate circulating tumor cell isolation from a patient's blood sample, 1mL of cells in suspension for electroporation were spiked into 20x diluted healthy whole blood (Zen-Bio, USA). The total number of cancer cells for each blood sample was spiked to be $5 \times 10^4$ cells. DPBS (Gibco®, Life technologies, USA) was used to dilute the sample to a final volume of 3 mL. Spiked blood samples were freshly prepared prior to each electroporation run to minimize interactions between patients' native WBC and spiked cancer cells.

### Cell purity assessment

To assess the purity of collected cells from blood-borne contaminants, HCC827 GR6 cells were labeled with Calcein AM following manufacturer protocol after trypsinization and spiked into the blood. After trapping and subsequent collection onto clear 96-well plates, cells were incubated in NucBlue following manufacturer protocol before centrifuging (180 g, 5 min) and resuspension in DPBS for whole-well fluorescent imaging (<1 hr). Cancer cells were identified as cells staining positively for both Calcein and NucBlue, WBCs were identified as cells

staining positively for NucBlue only, and RBCs were identified by their morphology from brightfield images.

## Statistical analysis

All statistical analyses and curve-fitting were conducted in Prism 8 (GraphPad, San Diego, CA) and Excel (Microsoft, Redmond, WA). Results are expressed as mean ± standard error of the mean (mean ± SEM). Comparisons of all experimental data sets were analyzed with one-way ANOVA followed by the Dunnett post-hoc test for multiple comparisons. A p-value $< 0.1$ was considered statistically significant for this study.

## Results and discussion

### Electroporation parameter optimization

Optimal electroporation conditions for the reversible permeabilization of HCC827 wt and GR6 were first identified using membrane-impermeable biomolecules prior to drug electroporation experiments. Molecular delivery of membrane-impermeant nucleus dyes (Propidium Iodide, 15 µM, Invitrogen and YOYO-1, 1 µM, Invitrogen) was used to visualize cytosolic delivery followed by Calcein AM incubation post-electroporation as a viability indicator (Calcein AM, 1 µM, Invitrogen and Calcein Red-Orange AM, 1 µM, Life Technologies) using fluorescent microscopy. Representative fluorescent images from (a) non-electroporated, (b) dead, and (c) electroporated cells are shown in Fig 2a–2c. Real-time visualization of membrane-impermeant molecule delivery into the cytosol was monitored using fluorescent microscopy (see S1 Movie).

From 2D fluorescent intensity maps (S2 Fig) generated for each voltage using computer-assisted image recognition code, respective electroporation efficiency, viability, and collected cell count of HCC827 and HCC827 GR6 at applied input voltages are shown in Fig 2d and 2e. HCC827 GR6 cells were electroporated with a wide input voltage distribution between 10V (0.6 kV/cm electric field), 15V (0.9 kV/cm), and 20V (1.2 kV/cm) to identify optimal electroporation conditions (Fig 2d). The optimal electroporation condition maximizes cytosolic

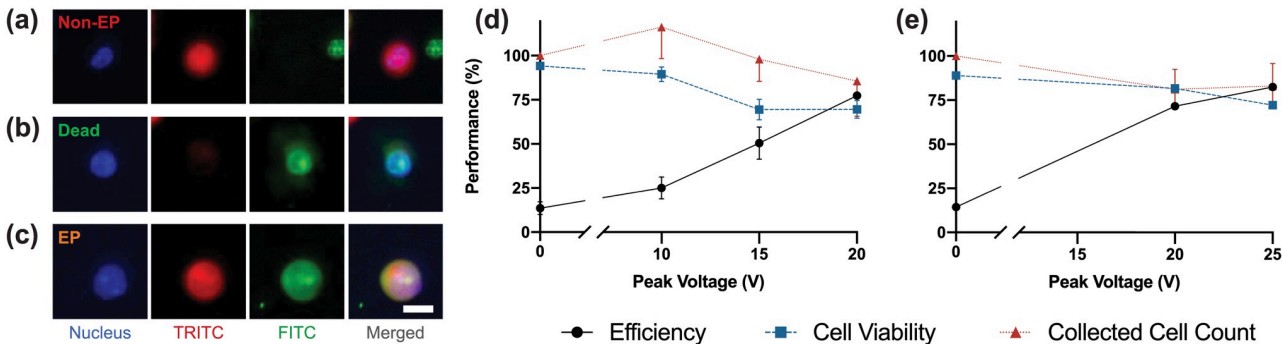

**Fig 2. Optimization of electroporation parameters for HCC827 GR6 and wt cells.** Microscopic images of HCC827 GR6 cells collected post-electroporation after intracellular delivery of membrane-impermeable fluorophore YOYO-1 (FITC) with Calcein Red-Orange AM (TRITC) counterstain as viability indicator. Cells were labeled with viable nucleus stain Hoechst-33342 (DAPI, Nucleus) prior to electroporation experiments. (a) Live cells that were not electroporated, indicated by the absence of YOYO-1 signal, (b) Dead cells lysed by the electroporation process characterized by the absence of Calcein Red-Orange AM signal, but uptake of YOYO-1, (c) Successful intracellular delivery of YOYO-1 to live cells, indicated by dual fluorescence of YOYO-1 and Calcein Red-Orange AM. Merged image of all 3 fluorescent channels is shown on the far right of the panel. Scale bars represent 25 µm. Efficiency (black), viability (blue), and collected cell count (red) were obtained for (d) HCC827 GR6 and (e) HCC827 wt cells. Error bars represent the standard error of the mean (n = 3, > 150 cells per experiment).

delivery while minimizing the amount of cell lysis and size of the bubble formation on the electrodes due to contributions from hydrolysis and joule heating. In the past, the optimum voltages have been previously demonstrated to vary between cell lines [32, 38]. Because the desired electroporation efficiency ($>$ 70%) of HCC827 GR6 cells was near the higher end of the applied voltages, HCC827 wt cells were electroporated at 20V and 25V (1.5 kV/cm) (Fig 2e). The optimal voltage for successful electroporation for both HCC827 wt and GR6 variant was determined to be 20V due to its high efficiency and consistency (77.4 ± 2.3% and 71.6% ± 0.6% for GR6 and wt, respectively), as well as high viability (69.5 ± 5.0% and 81.7% ± 1.3% for GR6 and wt, respectively). Side by side comparison of these values between the GR6 and wt cell lines at 20V can be seen in S3 Fig.

Cells trapped in vortex do not generally come into physical contact with the electrodes. However, cellular debris from cell bursting due to irreversible electroporation accumulates onto the electrodes. Both the magnitude of the rate at which debris build up and the generated bubble size increase proportionally with input voltage. While cell electroporation efficiencies were augmented at the upper limit voltages, these detrimental factors complicate cell trapping and thus the accurate assessment of electroporation performance. To mitigate these adverse effects, the voltage sequence was randomized across devices, and each voltage condition repeated once per device. While 20V input also generated bubbles, the magnitude of their size were smaller and thus were able to be quickly dissipated from continual flow into the device, minimizing impact on the cell trapping capacity of the vortex.

## Drug combination to sensitize drug-resistant cells

While tumors may respond to initial front-line therapy, selective pressure from prolonged drug exposure ultimately leads to the onset of drug resistance. One treatment modality that has seen developing interest and success in target drug resistance involves synergistic combinations of multiple licensed therapeutic agents [41, 42]. Combinatorial therapy has the potential to restore therapeutic efficacy in tumors with desensitized drug response. There are numerous examples in literature employing multi-agent therapy for treating drug-resistant breast cancer tumors, its success largely due to combinatorial strategies being able to target multiple sites [43–45]. Model studies involving both drug-sensitive and drug-refractory cancer cells are a critical intermediate step before screening CTCs from patient liquid biopsy samples. This study utilizes drug-sensitive and -resistant cell lines as surrogates for therapy refractive CTCs to validate viability response from combinatorial drug delivery. Gefitinib is well documented to exhibit a cytotoxic effect on HCC827 wt cells by blocking the EGFR kinase (IC50: $6.47^{e-3}$ μM). Prolonged exposure to gefitinib generates HCC827 GR6 clones harboring gefitinib resistance (IC50: 13.4 μM) (S4 Fig). A combination of gefitinib with PHA-665752, a c-MET inhibitor, restores gefitinib sensitivity in HCC827 GR6 cells [37].

Fig 3 shows the cell viability response over time in response to drug agents from both HCC827 GR6 and wt cells for electroporation and conventional well-plate control conditions. Cell density variation impacts from electroporation conditions were minimized by recording chemiluminescence intensities from two collective runs and precisely matching the length of time cells are kept in vortex. In addition, electroporation experiments involving single drug agents and vehicle controls were time-matched to combinatorial drug experiments involving multiple solution exchange steps by flowing wash buffer solution in place for the drug solution. Conventional well plate assays plated with roughly the average amount of cells to be collected from electroporation runs (~250 cells/well) were conducted in parallel to validate the performance of our sequential drug delivery system with its anticipated *in vitro* drug response. Drug concentration for the conventional well plate assays matched the total drug solute quantity

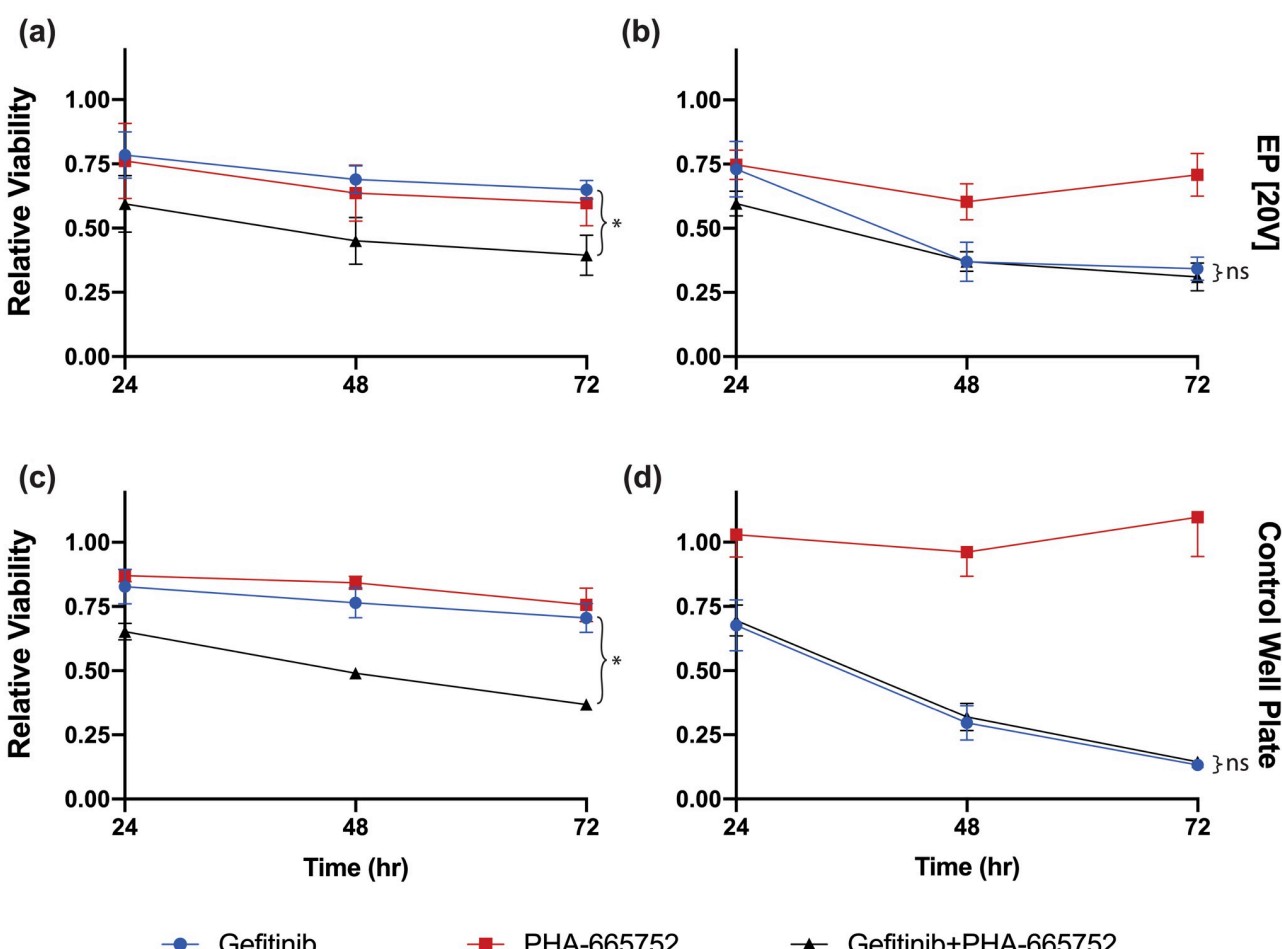

**Fig 3. Drug response curves of HCC827 cells to gefitinib and PHA-665752.** Viability response of (a) HCC827 GR6 and (b) wt cells after electroporation in gefitinib (100 nM), PHA-665752 (20 nM), or a sequential combination of both at time points 24, 48, and 72 hrs. Control conventional well-plate assays of (c) HCC827 GR6 and (d) wt cells, respectively, containing an identical solute quantity of drugs flown into the device during the electroporation step (100 and 20 pmol for gefitinib and PHA-665752, respectively). Shown viability% is normalized by the luminescence intensity of untreated control cells in their respective condition. Relatively high standard deviations associated with the collected cell counts can be attributed to repeated electroporation of cells on the same device, leading to eventual debris buildup on the electrodes, interfering with consistent cell capture. Error bars represent the standard error of the mean (n = 3). *p<0.1 vs gefitinib (control) groups.

flown through the device during the incubation step following electroporation (100 and 20 pmol for gefitinib and PHA-665752, respectively).

A 1:0.2 gefitinib:PHA-665752 drug combination ratio was identified to be optimal for our system due to consistent drug response from electroporated and well plate controls. Sequential electroporation-mediated delivery of gefitinib and PHA-665752 to both HCC827 GR6 and wt cells decreased cell viability and suggests the ability of the system to perform combinatorial drug sensitivity assessments (Fig 3a and 3b). In electroporation conditions, drug sensitivity to gefitinib in the wt cells is not observed with the GR6 cells. Electroporation-mediated delivery of PHA-665752 as a single agent does not elicit a substantial viability decrease in both cell lines, which is in good agreement with reported literature trends [37, 46, 47]. Compared to electroporation-mediated delivery, enhanced cytotoxic effects to gefitinib with sensitive wt cells could be attested to constant exposure to drugs in the well-plate controls over 72 hours. Interestingly, electroporation of PHA-665752 in wt cells showed a slight decrease in viability,

which could suggest that increased cellular influx of the drug-enhanced cytotoxic effects [48]. Well-plate controls of HCC827 GR6 and wt cells were conducted in parallel to validate electroporation experiments and exhibited synonymous drug response trends over time with electroporated cells for all drug conditions (Fig 3c and 3d).

## Cell purification and subsequent drug screening

CTCs are attractive as a routinely obtainable source of patient cells for clinical applications, as they reflect the real-time genomic state of the tumor [13, 49]. Purity challenges complicate downstream functional interrogations of CTCs, a lack of which also currently remains as a large roadblock for its clinical adaptation. Vortex-assisted cell purification systems allow for the facile capture and purification of large cancer cells with minimal contamination from smaller blood-borne components (red and white blood cells) [50]. Because their mechanism responsible for drug resistance is well-documented, HCC827 GR6 cancer cells were spiked into the blood to emulate drug-resistant CTCs as concept validation of our drug response assay workflow.

HCC827 GR6 cells spiked into whole diluted blood were processed for drug response screening using our system (S2 Movie). Experiments involving single-agent electroporation in PHA-665752 were omitted due to the negligible impact seen previously in viability response (Fig 3a). Fig 4a shows the purity, described as the ratio between RBC and WBC to cancer cells (94.6 ± 23.1 and 0.140 ± 0.057 for RBC and WBC, respectively), collected at the outlet. The impact from these blood cells onto cell viability readings was deemed negligible due to low occurrence. Control experiments for cells incubated in the drug solution incubation without electroporation were done in parallel (dashed lines). Fig 4b shows the cell viability with and without electroporation in response to drug agents for HCC827 GR6 cells spiked into the

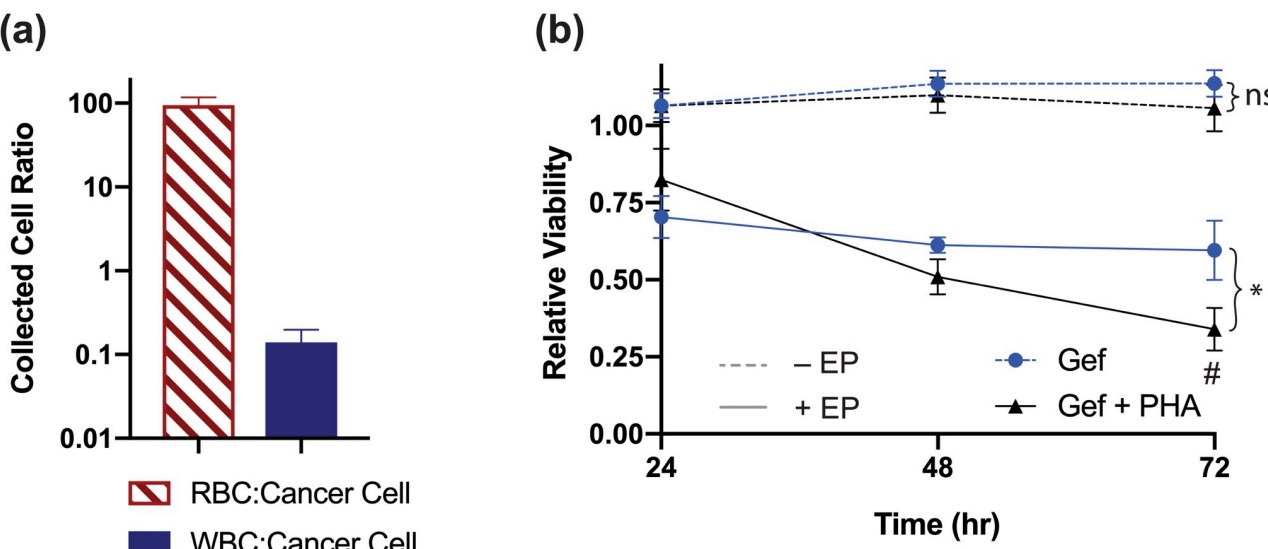

**Fig 4. Spiked blood sample purity assessment and drug response curve.** Spiked cancer cells (HCC827 GR6) were isolated from the blood and subsequently electroporated for drug combination testing. (a) Purity of isolated cancer cells (expressed in ratios relative to WBC and RBC counts). The number of contaminant blood-borne cells was deemed insignificant for interfering with luminescence assay. Error bars represent standard error of mean (n = 24) (b) Viability response of HCC827 GR6 after electroporation in gefitinib solution (100 nM) or a sequential combination of both (PHA-665752, 20 nM) at time points 24, 48, and 72 hrs (solid line). Control experiments without electroporation step during drug solution incubation are shown with dashed lines (100 and 20 pmol for gefitinib and PHA-665752, respectively). Shown viability% is normalized by the luminescence intensity of untreated (DPBS) control cells in their respective condition. Error bars represent the standard error of the mean (n = 3). *p<0.1 vs gefitinib (electroporation) groups. #p<0.1 vs gefitinib + PHA-665752 (- EP) groups.

blood. Incubation of HCC827 GR6 cells in drug solutions without electroporation for all conditions did not have an adverse effect on cell viability. Additionally, similar drug response trends are observed between electroporated cells, and well-plate controls over the course of 72 hours (Fig 3), suggesting that decreased viability in blood electroporation experiments mainly reflects drug combination mediated cytotoxicity.

The number of CTCs per mL of whole blood in cancer patients remains largely debated, as counts vary depending on the type and stage of cancer, as well as the CTC isolation technique used [51–53]. Regardless of the volume of liquid biopsy samples processed with our chip, vortex-based isolation techniques concentrate trapped cell populations into a small volume, and various downstream assays can be streamlined into the simple collection process. While it is true that using clinically relevant CTC concentrations found in blood would decrease the overall number of trapped cells using our system, future work should be dedicated towards rarer cell applications and integration with sensitive, single-cell based assays [54]. Calcein AM is a fluorescent-based viability indicator measuring intracellular esterase activity from individual cells, which is ideal for rare cell populations. Calcein AM is also a real-time indicator of intracellular oxidative activity, a critical physiological measurement implicated in tightly regulating cell proliferation, survival, and death [55–57]. As such, monitoring ratios of live and dead cells populations, as well as Calcein fluorescent intensities, have been qualitatively utilized to assess cytotoxic efficacies from prolonged (7 days) combinatorial Doxorubicin and aspirin treatment in cancer cell clusters [58]. Therefore, future studies applying Calcein intensity measurements from inherently rare patient CTCs, using a combination of both our workflow and fluorescent intensity map pipeline, as an alternative to assess viability response, are warranted.

## Conclusion

Current cancer treatment strategies are guided by the molecular expression and genomic status information derived from solid biopsies of the primary tumor. However, evolution of the molecular expression and genomic information from the tumor during disease progression necessitates longitudinal monitoring in order to guide appropriate treatment avenues. Liquid biopsies are an emerging alternative to solid biopsies because it entails a less-invasive sampling procedure that contains a more representative population of cells and analytes detailing the current tumor status. Despite their immediate relevance in clinical application, purification and analysis of these target analytes within liquid biopsies remain hurdles in their standardization and utility.

Detailed functional measurements from CTCs are essential for guiding clinical decisions regarding drug treatment modalities. However, CTC isolation from peripheral whole blood as well as *in situ* analysis represents an unmet need in order to identify effective therapeutic strategies [59]. Here, we present an integrated workflow that can assess therapeutic response from unaltered cancer cells spiked into blood, an essential first step towards profiling CTCs from patient samples. We systematically ascertained ideal electroporation parameters for cellular membrane permeabilization using on-chip electroporation visualization and automated cell-by-cell image classification sequence, a procedure that can be readily applied to optimize electroporation of native CTCs. Gefitinib-resistant cancer cells emulating drug-refractory CTCs in blood samples were purified with trace background contaminants and subsequently electroporated to deliver a combinatorial drug regime ablating resistance. Cancer cells processed within our system for drug viability assay reflected behavior as expected when compared to our conventional control experiments, suggesting the utility of electroporation as a delivery system and promising extension towards characterizing the response profiles from CTCs with high

fidelity and reproducibility. Development of this workflow for the clinic is critical towards realizing the promises functional liquid biopsy interrogation has towards longitudinal monitoring of disease treatment and relapse.

In conclusion, our proposed workflow demonstrates potential for translation into clinical and point of care settings for functional interrogations of CTCs obtained from patient samples. The versatility of our delivery system can incorporate therapeutic agents whose membrane diffusion or delivery efficiencies may not currently be well established, and could also facilitate the discovery of novel therapeutic combinations. We underscore the ability of our platform to retain the viability of cells throughout the drug electroporation process. This device has the potential to address the manufacturing of cell lines derived from CTCs, which has also been a critical hurdle due to cellular senescence and viability challenges [27]. Retaining viability is also critical for characterizing expression profiles and invasive behavior from CTCs demonstrating the most aggressive metastatic capacity [60]. Such studies may broaden our understanding of cancer biology through better elucidation and study of CTCs as the key driver in orchestrating the metastatic cascade.

## Supporting information

**S1 Movie. Real-time visualization of membrane-impermeable molecule delivery.** Initial trapped cells are visualized by NucBlue staining (Hoechst 33342, DAPI). After electroporation pulses, membrane-impermeable molecule (YOYO-1, FITC) delivery into the cytosol can be monitored using fluorescent microscopy in real time, allowing for rapid electroporation parameter adjustments.
(MP4)

**S2 Movie. Blood purification and subsequent drug electroporation.** Drug-resistant cancer cells spiked into the blood can be processed using inertial microfluidics mediated vortex separation. Shortly after purification, trapped cancer cells are electroporated for integrated drug response assays.
(MP4)

**S1 Fig. Representative microscopic images of the electroporation sequence.** The white dashed lines outline the cell trapping chambers. Cancer cells (HCC827 GR6), pre-stained with live-cell nucleus marker Hoechst 33342, are isolated into cell-trapping microvortices. Cytosolic delivery via electroporation is visualized by detecting fluorescent signals of a membrane-impermeable dye Propidium Iodide (PI, 15 μM) from trapped cells. Fluorescent images of cells incubated in PI solution are taken before (Molecular Incubation) and after (Molecular Delivery) electroporation. Dead cells uptake PI prior to electroporation (red box). Successful electroporation is indicated by the overlap of PI and nucleus marker fluorescent signals (purple box). Cells that are not electroporated are indicated by the absence of PI fluorescent signals after electroporation (blue box). Scale bar = 200 μm.
(TIF)

**S2 Fig.** Generated 2D fluorescent intensity maps from each collected cell after electroporation in (a) 0V, (b) 10V, (c) 15V, and (d) 20V. Thresholding for determining successful electroporation (e) and cell viability (f) are described in the materials and methods section. Thresholding divides the intensity maps into 4 distinct quadrants, where the Calcein +/YOYO-1—quadrant represents viable but un-electroporated cells, Calcein -/YOYO-1 + quadrant represents dead cells presumably burst by the electroporation procedure, and Calcein +/YOYO-1 + quadrant represents viable and successfully electroporated cells. Calcein -/YOYO-1—quadrant represents cells that have fluorescent intensities too low to be

considered for electroporation efficiency assessment. n > 150 cells for each intensity map from experiments in triplicates.

(TIF)

**S3 Fig. Side-by-side electroporation efficiency assessment for HCC827 cell line at 20V.** Efficiency (a, black), viability (b, blue), and collected cell count (c, red) are reported for GR6 (dark) and wt (light). The optimal voltage for successful electroporation for both HCC827 wt and GR6 variant was determined to be 20V due to consistent performance. Error bars represent standard error of mean (n = 3).

(TIF)

**S4 Fig. Dose-response curves from (a) HCC827 GR6 and (b) HCC827 wt in response to gefitinib.** Cells (4000 cells/well) in conventional opaque assay well plates were treated with gefitinib at the indicated concentrations and measured for viability after 72 hours using the RealTime-Glo™ MT Cell Viability Assay Kit. Viability % is shown relative to untreated (DMSO) control wells. IC50 values were determined by using nonlinear regression analysis (variable slope, 4 parameter) and determined as the concentration where percent inhibition of cell viability relative to vehicle (DMSO) well is equal to 50%. Error bars represent standard error of mean (n = 3).

(TIF)

**S5 Fig.** (a) HCC827 GR6 cells per well were plated on conventional 96-well assay plates and monitored for change in luminescence over time (0, 24, 48, 72 hrs). (b) Fold-change in luminescence linearly correlates over time, suggesting that the overall fold-change impact from various starting cell counts is negligible. A lower fold-change in luminescence, especially in low cell counts, could be attributed to difficulties in measuring low signals. Error bars represent standard error of mean (n = 3).

(TIF)

## Acknowledgments

We would like to express our gratitude to all healthy whole blood donors who contributed to this study. The authors would like to thank Dr. Pasi A. Jänne's group for kindly providing HCC827 cell lines, Huy Vo, and the Whiting School of Engineering Whitaker Microfabrication Lab, as well as Mark Lecates, and the Maryland NanoCenter and its FabLab, for support with photolithography and device fabrication. Fig 1a created with BioRender.com.

## Author Contributions

**Conceptualization:** Hyun Woo Sung, Sung-Eun Choi, Chris H. Chu, Mengxing Ouyang, Soojung Claire Hur.

**Data curation:** Hyun Woo Sung, Sung-Eun Choi, Chris H. Chu, Mengxing Ouyang, Soojung Claire Hur.

**Formal analysis:** Hyun Woo Sung.

**Funding acquisition:** Sung-Eun Choi, Soojung Claire Hur.

**Investigation:** Hyun Woo Sung, Soojung Claire Hur.

**Methodology:** Mengxing Ouyang, Srivathsan Kalyan, Nathan Scott, Soojung Claire Hur.

**Project administration:** Soojung Claire Hur.

 

**Resources:** Soojung Claire Hur.

**Software:** Hyun Woo Sung, Sung-Eun Choi, Mengxing Ouyang, Soojung Claire Hur.

**Supervision:** Sung-Eun Choi, Soojung Claire Hur.

**Validation:** Hyun Woo Sung, Sung-Eun Choi.

**Visualization:** Hyun Woo Sung, Sung-Eun Choi, Mengxing Ouyang, Soojung Claire Hur.

**Writing – original draft:** Hyun Woo Sung.

**Writing – review & editing:** Hyun Woo Sung, Sung-Eun Choi, Chris H. Chu, Mengxing Ouyang, Soojung Claire Hur.

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
