## [Decision Letter · Decision Letter 0]

7 Jan 2022

PONE-D-21-30672Sensitizing Drug-Resistant Cancer Cells from Blood using Microfluidic ElectroporatorPLOS ONE

Dear Dr. Hur,

Thank you for submitting your manuscript to PLOS ONE. After careful consideration, we feel that it has merit but does not fully meet PLOS ONE’s publication criteria as it currently stands. Therefore, we invite you to submit a revised version of the manuscript that addresses the points raised during the review process.

We look forward to receiving your revised manuscript.

Kind regards,

Arum Han, Ph.D.

Academic Editor

PLOS ONE

Journal Requirements:

S.C.H. benefits financially from royalty payments from the Vortex Biosciences, Inc.

Reviewers' comments:

Reviewer's Responses to Questions

**Comments to the Author**

1. Is the manuscript technically sound, and do the data support the conclusions?

Reviewer #1: Yes

2. Has the statistical analysis been performed appropriately and rigorously? 

Reviewer #1: Yes

3. Have the authors made all data underlying the findings in their manuscript fully available?

Reviewer #1: Yes

4. Is the manuscript presented in an intelligible fashion and written in standard English?

Reviewer #1: Yes

5. Review Comments to the Author

Reviewer #1: Please see the attached comments for details. Briefly, the manuscript described a microfluidic systems integrated with electrodes for trapping and electroplating cells. The experiments were designed and executed well.

6. PLOS authors have the option to publish the peer review history of their article (what does this mean?). If published, this will include your full peer review and any attached files.

Reviewer #1: No

---

## [Author Response · Author response to Decision Letter 0]

18 Feb 2022

We thank the reviewer for their constructive comments and suggestions and have made the necessary edits to address their concerns. Please see the following point-by-point address.

Comment #1: I have the impression that this study aims to develop a workflow that could allow direct assessment of patient samples. However, the study works with blood samples that are spiked with HCC827 wt and HCC827 GR6 cells. For proof-of-concept, spiking cells with the blood sample is reasonable. It may be better to make it clear that the workflow has not been able to work with patient sample directly yet and point out this would be the next step work.

• This is a great point raised by the reviewer. The current study focuses on spiked cell experiments to develop the electroporation workflow for future transition into utilizing patient blood samples. The authors would like to respectfully point out that vortex technology has been demonstrated to capture CTCs, and that the electroporation workflow utilizing spiked cells has been stated in the abstract (Sentence 7 “HCC827 GR6 cells spiked…”), introduction (Paragraph 4 Sentence 8 “… we detail a procedure…”), results and discussion (Section: CTC purification and subsequent drug screening, Paragraph 2 Sentence 1), and conclusion (Paragraph 2 Sentence 5 “Gefitinib-resistant cancer cells emulating drug-refractory CTCs…”) to avoid confusion. The authors agree with the comment, and both the limitation of this current study and future goals to transition into patient samples have been more clearly stated in the updated manuscript (both introduction and conclusion). 

Comment #2: It is not clear whether or not flushing the device with wash buffer prior to cell collection affects the processed cells, for example, are the cells flushed away? 

• The authors appreciate the reviewer’s comment. When transitioning between any molecule/drug/wash buffer solution, our solution exchange system utilizes a simultaneous co-flow transition (i.e., there is no abrupt on-off switching from one vial to another) of solution vials at 40 psi to ensure that the flow rate through the device, at any given point when switching solutions, is sufficient to retain cell trapping. Similar solution exchange protocols have been demonstrated numerous times for cell trapping in the past (Ouyang et al. Sci Rep 2017, Vickers et al. Anal Chem 2014, Yun et al. Lab Chip 2013). This information has been revised on page 7 for improved reading clarity in the updated manuscript. 

Comment #3: When collecting the cells to 96-well plates, was the cell density controlled in each well? If yes, how? If not, would the cell density variation affects the study? 

• We thank the reviewer for this important comment. While the exact processed cell density cannot be ascertained until the endpoint, we have identified that the number of collected cells is largely dependent on the flow condition of the device as it directly affects robust vortex formation for cell trapping. The effects of these conditions on the cell density per well were minimized by closely monitoring the condition of the device and randomizing the input voltage sequence for electroporation. Additionally, the trapping efficiency and input cell concentrations were closely monitored, and the solution exchange procedure purposely designed to retain trapped cell populations. For viability assays that measure the reducing potential of all viable cells present, impacts from cell density variations were mitigated by recording chemiluminescence readings from two collective runs. Fold-change in cell chemiluminescence from each well was calculated in comparison over relative change to lessen the effects of cellular density variations. In the supplementary information provided with the manuscript, we observed that cell density variation effects on fold-change in cell chemiluminescence were negligible. The manuscript has been revised to include these details. 

Comment #4: It is observed that 25V cause bubbles in addition to other problems. I believe 20V would cause bubble as well although it might not affect the study significantly. Please comment on that. 

• This is an important observation pointed out by the reviewer. In the video file “S2_Movie.mp4”, the bubble formation at the electrodes, due to contributions from both electrolysis of the ionic solution and joule heating effects, can be observed in brightfield. Without sacrificing too much electroporation performance, the magnitude of the bubbles and rate at which cellular debris from burst cells accumulate on the electrodes were lower when operating at 20V compared to 25V when electroporating HCC827 wt cells. Additionally, bubble formation at 20V does not significantly impede the trapping capacity of the vortices and are also rapidly dissipated away from continual flow into the device (i.e., the large majority of cells in the view remain uninterrupted and remain robustly trapped in vortex after electroporation sequence). These explanations and details will be included in the revised manuscript. 

Comment #5: It seems the liquid sample with spiked cells is directly in contact with the electrodes. If this is correct, would the cells adhere to the electrodes quickly leading to unstable performance? Please comment whether or not a passivation layer designed on top of the electrodes to separate the liquid sample from the electrodes is possible to make the system work?

• We thank the reviewer for this insightful suggestion. As stated before, cellular debris naturally accumulates onto the charged electrodes over the course of electroporation. In the past, we have tested passivating our microfluidic devices with 5% w/v Pluronics F-127 in DPBS to minimize biofouling, but this yielded insignificant impact in deterring cell debris adhesion (data not shown). There are several works with microscale interdigitated electrode circuits utilizing thin passivation layers to physically eliminate bubble generation and cell contact between electrodes (Jungreuthmayer et al. Lab Chip 2011, Pandian et al. Biomicrofluidics 2020). These passivation layers are integral for these systems where flow is restricted, cells are in physical contact with the underlying electrodes, or bubbles directly interfere with sample collection procedures. Additionally, the relative permittivity of a passivation layer compared to liquid ionic solution suggest significantly higher voltage inputs in order to generate the required electric fields for reversible membrane permeabilization, and thus may additionally require a sophisticated microelectrode fabrication technique to follow. Because the cells are in vortex orbit and the observed bubbles in our system are dispersed rapidly, similar working performance with these mentioned systems can be achieved with a simpler fabrication process. 

Comment #6: I suggest “a vacuum environment of 6e-6 torr” change to “a vacuum environment of 6e-6 torr”

• This suggestion has been included in the revised manuscript.

Thank you for your time, consideration, and comments to our initial submission. We hope that the following response and manuscript amendments meet the publishing criteria set forth by the editors, reviewers, and PLOS One.

---

## [Editor Report · Decision Letter 1]

22 Feb 2022

Sensitizing drug-resistant cancer cells from blood using microfluidic electroporator

PONE-D-21-30672R1

Dear Dr. Hur,

We’re pleased to inform you that your manuscript has been judged scientifically suitable for publication and will be formally accepted for publication once it meets all outstanding technical requirements.

Kind regards,

Arum Han, Ph.D.

Academic Editor

PLOS ONE
---

## [Editor Report · Acceptance letter]

28 Feb 2022

PONE-D-21-30672R1 

Sensitizing drug-resistant cancer cells from blood using microfluidic electroporator

Dear Dr. Hur:

I'm pleased to inform you that your manuscript has been deemed suitable for publication in PLOS ONE. Congratulations! Your manuscript is now with our production department. 

Kind regards, 

on behalf of

Dr. Arum Han 

Academic Editor

PLOS ONE